

# Brief communication: Surface energy balance differences over Greenland between ERA5 and ERA-Interim

Uta Krebs-Kanzow[1], Christian B. Rodehacke[1,2], and Gerrit Lohmann[1,3]

[1]Alfred-Wegener-Institut, Helmholtz-Zentrum für Polar- und Meeresforschung, Bremerhaven, Germany
[2]Danish Meteorological Institute, Copenhagen, Denmark
[3]University of Bremen, Bremen, Germany

**Correspondence:** Uta Krebs-Kanzow (uta.krebs-kanzow@awi.de)

**Abstract.** We compare the main atmospheric drivers of the melt season over the Greenland Ice Sheet (GrIS) in ERA5 and ERA-Interim (ERAI) in their overlapping period 1979–2018. In summer, ERA5 differs significantly from ERAI, especially in the melt regions: averaged over the lower parts of the GrIS, mean near-surface temperature is $1\,\mathrm{K}$ lower, while the mean downward shortwave radiation at the surface is on average $15\,\mathrm{Wm}^{-2}$ higher than in ERAI. Comparison with observational weather station data shows a significant warm bias in ERAI and for ERA5 a significant positive bias in downward shortwave radiation. Consequently, methods that previously estimated the GrIS surface mass balance from the ERAI surface energy balance need to be carefully recalibrated before converting to ERA5 forcing.

## 1 Introduction

Greenland summer temperatures are experiencing a persistent warming trend. In coastal instrumental temperature records, Hanna et al. (2021) diagnose a significant $1.7\,\mathrm{K}$ increase from 1991 to 2019, and in SSP5-8.5 projections, a mean warming of $5.3\,\mathrm{K}$ is projected for the 21st century. The associated reduction in Greenland Ice Sheet's surface mass balance (SMB) leads to more runoff that ultimately raises the global sea level. According to the conservative estimate of Hanna et al. (2021), the equivalent sea level rise amounts to more than $10\,\mathrm{cm}$ by the end of this century. Surface mass balance models (EBMs hereafter), such as BESSI (Born et al., 2019) or dEBM (Krebs-Kanzow et al., 2021), represent the key physical processes that determine the surface mass balance, and they can be used to directly infer changes in SMB from basic surface climate variables which typically include surface downward shortwave and longwave radiation, near-surface temperature, and precipitation.

EBMs provide a low-cost alternative to computationally intensive regional climate model simulations to downscale the SMB and reproduce the narrow ablation zone along the lower elevated ice sheet margins. In a model intercomparison (Fettweis et al., 2020), EBMs were shown to be able to reconstruct the 1979–2012 SMB of the GrIS from relatively coarse-resolution ERA-Interim climate reanalysis, even though EBMs proved to be somewhat less skillful than regional climate models (RCMs), which may be partly related to the relatively coarse resolution of the ERA-Interim forcing (approximately $79\,\mathrm{km}$) compared to the higher resolution of participating RCMs of up to $5.5\,\mathrm{km}$. However, ERA-Interim was suspended in 2018 and is replaced by the ongoing ERA5 reanalysis product, which provides higher horizontal resolution (approximately $30\,\mathrm{km}$) and dates further





back to 1959. Thus, the potential and relevance of EBMs have increased when used in combination with this higher-resolution
climate forcing now available.

It is, therefore, desirable to update current SMB simulations to ERA5 forcing and, where necessary, to adjust existing EBM
parameters, which will require an assessment of the differences between the two data products due to changes in the near-
surface radiation scheme, cloud scheme, or surface boundary layer. To this end, we compare key climate properties between
ERA5 and ERA-Interim to complement previous comparisons (Wang et al., 2019; King et al., 2022; Delhasse et al., 2020)
focussing on the Greenland (summer) surface energy balance.

## 2 Data and method

We use two global atmospheric reanalysis datasets produced by the European Centre for Medium-Range Weather Forecasts
(ECMWF). The Reanalysis Era-Interim (ERAI) covers the period from January 1979 to August 2019 and has a spatial resolu-
tion of about $80\,\mathrm{km}$ over Greenland. The more recent Reanalysis v5 (ERA5) begins in January 1959, runs until the present, and
has a finer resolution of about $30\,\mathrm{km}$. We compare climate properties that primarily control surface ablation over the Greenland
Ice Sheet (GrIS) for those years entirely covered by both the ERA-Interim and ERA5 data sets, namely the joint period 1979–
2018. Specifically, we compare the 2m-air temperature (T2M), downward shortwave radiation at the surface (SWD), effective
atmospheric emissivity ($\epsilon$), and cloud cover (CC).

The analysis focuses on the summer months (June, July, and August; hereafter JJA) and the lower parts of the ice sheets
between sea level and $2000\,\mathrm{m}$ elevation. The effective atmospheric emissivity $\epsilon$ is derived from the downward longwave
radiation (LWD) at the surface and $T2M$ according to the Stefan-Boltzmann-Law:

$$\epsilon = \frac{LWD}{\sigma\, T2M_{int}^4} \tag{1}$$

where $\sigma$ is the Stefan-Boltzmann constant.

For these variables, we analyse mean differences (Fig. 1) and standard deviations (Fig. S1 in the supplement) of correspond-
ing ERA5 - ERAI differences for the summers in the 1979–2018 overlap period. A corresponding comparison of the annual
fields can be found in the supplement (Fig. S2, S3). All climate variables considered were bilinearly interpolated to the $1\,\mathrm{km}$
grid used by the Ice Sheet Model Intercomparison Project for CMIP6 (ISMIP6); (Note that effective emissivity is calculated
from the coarse resolution LWD and T2M and then interpolated).

We also scale temperatures with respect to a common orography, the $1\,\mathrm{km}$ ISMIP6 orography $H_{ice}$ (Morlighem et a., 2014),
to reduce those temperature differences which are related to differences in topography between both reanalyses (resulting from
the higher horizontal resolution in ERA5). To this end, a lapse rate of $\gamma = -0.005\,\mathrm{K\,m^{-1}}$ is applied to adjust ERA5 and ERAI
2m-air temperatures:

$$T2M = T2M_{int} + \gamma(H_{ice} - H_{int}) \tag{2}$$

where $T2M_{int}$ and $H_{int}$ are the interpolated near-surface temperature and surface elevation from the respective reanalysis data
sets (i.e., ERA5 or ERA-Interim), respectively. The applied lapse rate is at the low end of summer lapse rates over Greenland





slopes estimated from ERA5 and ERA-Interim 2m-air temperatures, which typically vary between $-5\,\mathrm{K\,km^{-1}}$ and $-7\,\mathrm{K\,km^{-1}}$ (Fig. S4) in agreement with climate simulations (Erokhina et al., 2017). A comparison of the mean T2M ERA5-ERAI biases for different lapse rate choices ($0\,\mathrm{K\,km^{-1}}$, $-5\,\mathrm{K\,km^{-1}}$, $-7\,\mathrm{K\,km^{-1}}$, $-10\,\mathrm{K\,km^{-1}}$) is provided in the supplement (Fig. S5, S6)

To also compare the reanalysis data to observational data, we bilinearly interpolate T2M, SWD, and emissivity ($\epsilon$) from the $1\,\mathrm{km}$ grid to locations of automatic weather stations (AWS) from the PROMICE network (Fausto et al., 2021; Ahlstrom et al., 2008). We consistently apply a lapse rate correction to downscale T2M to the altitude of the weather stations and compare it to monthly mean near-surface temperature measurements. A comparison with uncorrected temperatures is given in the supplement (Fig. S7). In addition, we compare the corresponding reanalysis data with monthly mean downward shortwave

radiation observations and effective emissivities calculated from in situ longwave radiation and near-surface temperature measurements. This comparison with observational data is similar, but not identical, to parts of Delhasse et al. (2020) because we use a different interpolation strategy and apply a lapse rate correction to account for altitude differences.

## 3    Differences between ERA5 and ERAI in summer

During the summer months, ERA5 and ERAI exhibit pronounced differences in the variables considered (Fig. 1). Over the

entire 40-year period, mean summer 2m-air temperatures (T2M) in ERA5 are more than $1\,^{\circ}\mathrm{C}$ colder than in ERAI over most parts of the ice sheet except the South Eastern margins and the southern dome region. The mean bias exceeds two standard deviations of the interannual variability almost everywhere north of $66°\mathrm{N}$. The applied lapse rate correction of $-5\,\mathrm{K\,km^{-1}}$ appears to be well chosen as the spatial difference between the two reanalysis products increases when no lapse rate correction is applied, or a higher lapse rate of $-10\,\mathrm{K\,km^{-1}}$ is chosen (Fig. S5).

The summer shortwave downward radiation at the surface (SWD) is stronger in ERA5 than in ERAI over the main ice sheet. The mean bias exceeds two standard deviations on the lower parts of the ice sheet where SWD in ERA5 exceeds SWD in ERAI by approximately $15\,\mathrm{W\,m^{-2}}$. Only isolated ice caps and the outermost margins of the main ice sheet show a distortion of the opposite sign in some places. In contrast, the emissivity in ERA5 deviates only slightly from ERA-Interim in most parts of the ice sheets except for a pronounced negative bias at the central-eastern margins (here the mean bias exceeds two standard

deviations between $66\,^{\circ}\mathrm{N}$ and $70\,^{\circ}\mathrm{N}$). The lower parts of the ice sheet have mostly lower emissivity values in ERA5, which is consistent with the pronounced positive shortwave radiation bias in ERA5. However, these features are not accompanied by a correspondingly lower cloud cover in ERA5 (Fig. 1).

A comparison with automatic weather station (AWS) measurements from the PROMICE network (Fausto et al., 2021; Ahlstrom et al., 2008) (Fig. 2) shows no significant bias in the ERA5 temperatures while the ERAI temperatures are signifi-

cantly warmer than the observations with a mean bias of $0.74\,\mathrm{K}$ (according to t-tests with a 0.05 significance level). Applying the lapse rate correction of $5\,\mathrm{K\,km^{-1}}$, for example, reduces the spread of the reanalysis data around the observational data considerably but also reinforces the warm bias in ERAI (Fig. S7). This comparison also shows a larger scatter in SWD around



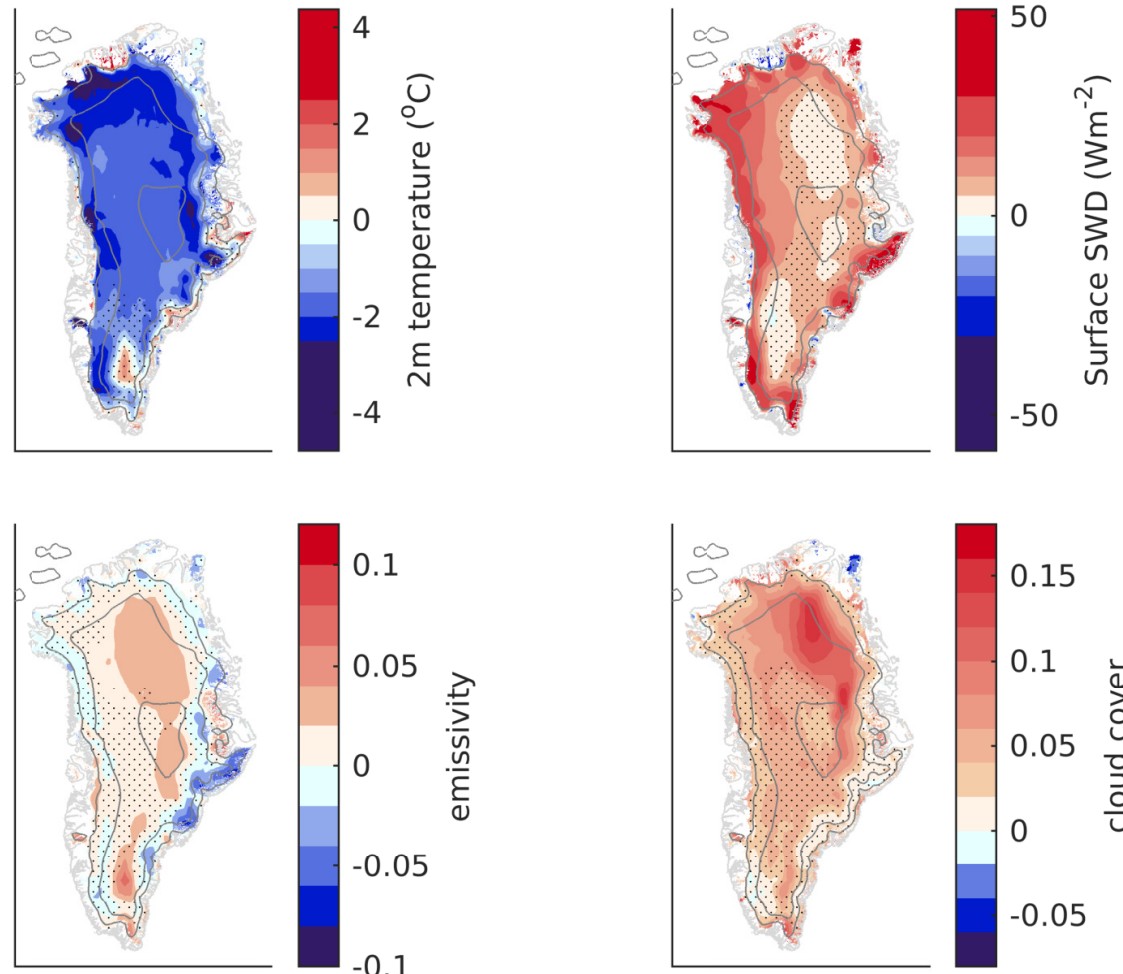

**Figure 1.** Mean bias between ERA5 and ERAI for the summer mean (i.e., June, July, and August, JJA) 1979–2018 period of the 2m-air temperature (top left), downward shortwave radiation (top right), emissivity (bottom left) and cloud cover (bottom right). Stippling indicates regions where the mean bias is smaller than two respective standard deviations.

observed station data, a significant positive bias in SWD for ERA5, and a significant negative bias in emissivity for both reanalyses. There is no significant bias in the ERAI SWD data.

Over the lower parts of the ice sheet ($< 2000\,\mathrm{m}$), differences are pronounced, as shown by the temporal evolution of the fields considered (Fig. 3). This height range covers the ablation zone which is generally limited to altitudes below $2000\,\mathrm{m}$ above sea level. The averaged ERA5-ERAI 2m-air temperature difference here varies around a mean of $-1.0\,\mathrm{K}$ with a standard deviation of $0.24\,\mathrm{K}$ (Fig. 3). This bias is enhanced by $25\,\%$ during the period between 2002–2009 when it is consistently above $-1.25\,°\mathrm{C}$.



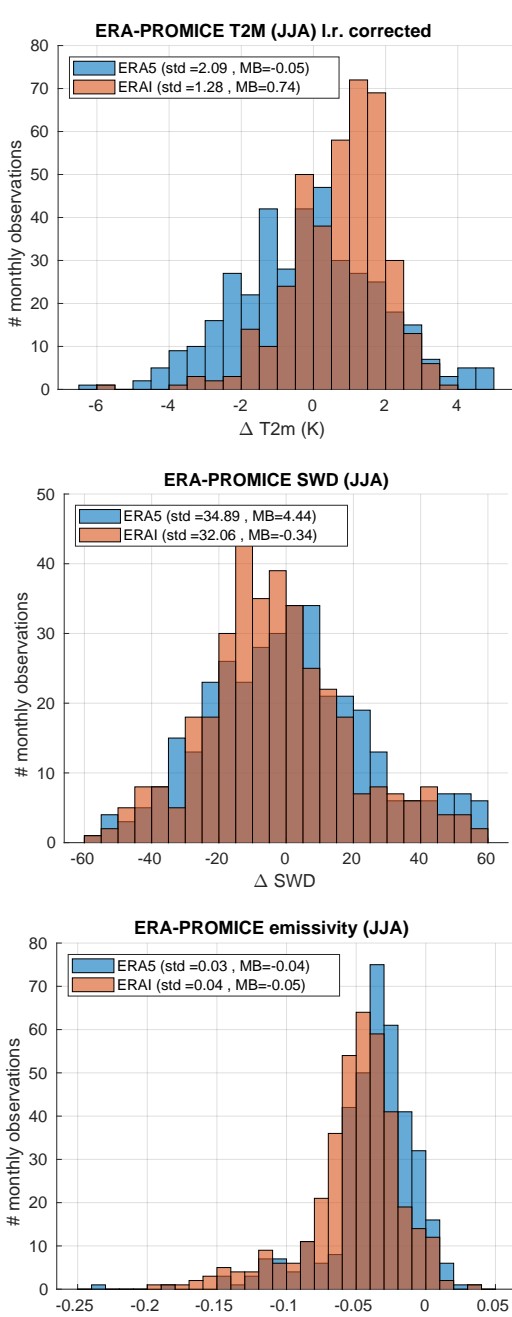

**Figure 2.** Distribution of ERA5 and ERAI biases with respect to monthly PROMICE observations for the summer months (June, July, August) in 2007–2016: 2m-air temperature (top), downward shortwave radiation (center), and emissivity (bottom). The text box insets provide standard deviation (std) and mean biases (MB) for the respective distributions.





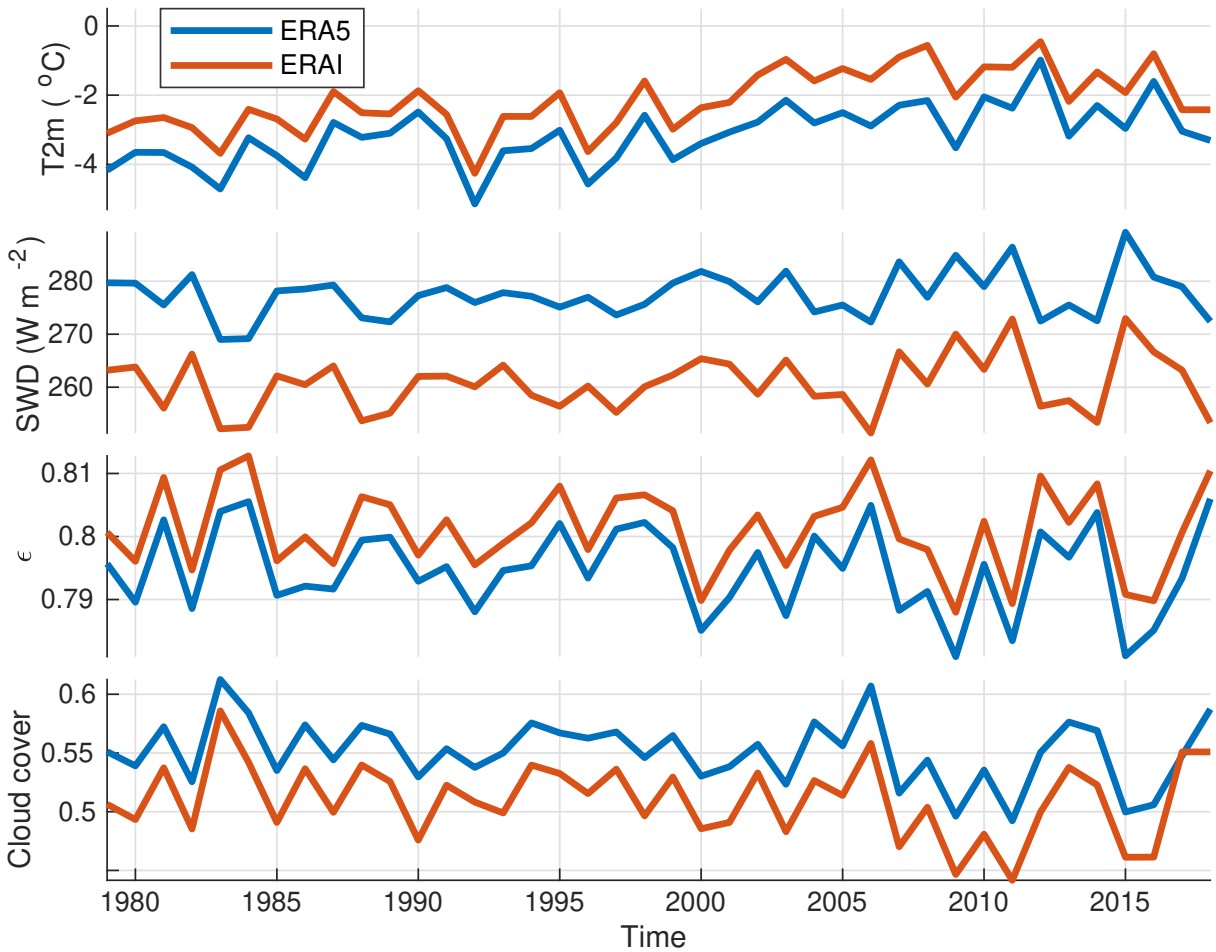

**Figure 3.** 1979–2016 yearly summer means of (a) 2m-air temperature, (b) downward surface shortwave radiation, (c) emissivity, and (d) cloud cover for ERA5 (blue) and ERAI (red) averaged over the $0\,\mathrm{m}$ to $2000\,\mathrm{m}$ elevation range of the GrIS.

This colder period in ERA5 may be partly related to a known cold bias of ERA5 in the lower stratosphere between 2002 and
2006 (one reason why ECMWF released ERA5.1, a rerun of this period Simmons et al. (2020)). The SWD bias varies around
16.7, $\mathrm{W\,m^{-1}}$ with a standard deviation of 1.6, $\mathrm{W\,m^{-1}}$.

Bias in cloud cover appears to be about $5\,\%$ throughout the period 1979–2018, with the remarkable exception of the year
2017. In 2017, both, ERA5 and ERAI show similar mean cloud cover values over the lower ice sheet. Differences in effective
emissivity also remain largely stable over time, with a mean bias of -0.0063 and a standard deviation of 0.0017.

These biases in the ablation zone imply that using either ERAI or ERA5 climate to drive the same energy balance model
will result in different surface melt rate distributions in GrIS. We use the simplified formulation of the surface energy balance



for a melting snow surface as given in Krebs-Kanzow et al. (2018) to estimate the resulting melt rate difference as about

$$\Delta M = ((1 - A) \, \Delta SWD + k_1 \, \Delta T2M) \; \frac{1}{\rho \, L_f}, \tag{3}$$

where $A$ is the albedo of the surface, $\rho = 1000 \, \mathrm{kg \, m^{-3}}$ is density of water, $L_f = 3.34 \times 10^5 \, \mathrm{J \, kg^{-1}}$ is latent heat of fusion,
and the parameter $k_1$ is chosen to be $10 \, \mathrm{W \, K^{-1} \, m^{-2}}$. The differences between ERA5 and ERAI of $\Delta SWD = 15 \, \mathrm{W \, m^{-2}}$
and $\Delta T2M = -1 \, \mathrm{K}$ yield a range from $\Delta M = -0.25 \, \mathrm{mm \, day^{-1}}$ for a low albedo of $A = 0.4$ in the dark bare ice zone to
$-2 \, \mathrm{mm \, day^{-1}}$ for a fresh snow albedo of $A = 0.9$. Therefore, melt rates from ERA5 are expected to remain mostly lower than
the respective estimates based on ERAI, especially at higher altitudes where albedo is generally high. This would result in a
lower equilibrium line and stronger melt gradients between the equilibrium line and the ice sheet's margin. However, stronger
SWD differences may overcompensate for the colder temperatures in the darker parts of the ablation zone and consequently
lead to stronger melt estimates under ERA5 forcing.

## 4 Conclusions

Our comparison reveals substantial and temporally coherent differences between ERA5 and ERA-Interim, resulting in a modi-
fied surface energy balance over the GrIS. ERA5 is characterized by systematically colder near-surface temperatures and more
intense insolation in summer. The difference in shortwave radiation downward (SWD) is particularly pronounced along the
lower parts of the ice sheets where higher spatial resolution of ERA5 better represents the steep orography.

Correcting the near-surface temperatures with a lapse rate of $-5 \, \mathrm{K \, km^{-1}}$ reduces the differences between the two reanalysis
products and improves the comparison with monthly observations from PROMICE weather stations for both data sets too. This
result is consistent with slope lapse rates diagnosed from both data sets and stresses the benefit of this simple downscaling
method when dealing with coarse-resolution temperature fields.

In contrast to Delhasse et al. (2020), we find a significant warm bias of ERAI relative to weather station data, but this is only
fully evident when a lapse rate correction is applied, while SWD appears to be slightly overestimated in ERA5.

The observed differences between ERA5 and ERAI have implications for the estimation of surface melt and ultimately the
release of runoff. Replacing ERAI with ERA5 forcing in an energy balance model of the GrIS may therefore require some
re-calibration to reproduce existing observations (e.g., IMBIE Team, 2020).

*Code and data availability.* The reanalysis data sets are provided by the European Centre for Medium-Range Weather Forecasts (ECMWF).
Information about ERA-Interim and ERA5 are given at https://www.ecmwf.int/en/forecasts/dataset/ecmwf-reanalysis-interim [accessed 2021-
11-12] and https://www.ecmwf.int/en/forecasts/dataset/ecmwf-reanalysis-v5 [accessed 2021-11-12]

.



*Author contributions.* UKK performed the analysis and lead the manuscript writing. All authors contributed to the interpretation of the results and proofreading of the manuscript.

*Competing interests.* The authors declare that they have no conflict of interest.

*Acknowledgements.* U. Krebs-Kanzow acknowledges the Helmholtz Climate Initiative REKLIM (Regional Climate Change) and the research program PoF IV "Changing Earth – Sustaining our Future" of the Alfred Wegener Institute. C. Rodehacke has received funding from

the European Union's Horizon 2021 research and innovation programme under grant agreement No 101056939 for the project RESCUE, and he has also received funding via the Alfred Wegener Institute's PACES2 research program.



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
