# Peer review of "Brief communication: Surface energy balance differences over Greenland between ERA5 and ERA-Interim"

_EGUsphere, 2023_

## Referee Comment (RC2)

**A Review of «** *Brief communication: Surface energy balance differences over Greenland between ERA5 and ERA-Interim* **» by Krebs-Kanzow et al.**

*Overview and general comment*

Authors present here a comparison of a part of surface energy balance components between both ERA-Interim and ERA5 reanalysis with the aim to adapt SMB calculation by EBMs to the highlighted differences. Originality of the comparison come from the focus on the area below 2000m of the ice sheet to better compare the datasets over the ablation area, and the use of a temperature lapse rate to correct the 2m-temperature of differences in surface elevation when interpolated on a common 1km-grid. The comparison is clear, straightforward and well-written.

*Major Comments*

A complete analyse of the surface energy budget components (longwave radiation too) should be presented, at least in supplements if it doesn't add significant conclusions.

Emissivity is calculated on the respective grid of both reanalysis, which implies that when downscale to 1km-grid, there is no correction relative to the elevation differences whereas ε is depending on the temperature. It would help to have an idea of this influence as you are considering a lapse rate to correct the temperature.

To estimate different lapse rates to correct temperature of surface elevation differences, authors calculate local lapse rates for each grid. Why don't use directly these lapse rates to correct temperature? Depending of the results, this could be add in the comparison (Figure S5) in the supplements.

AWS data in ablation area are used to compare both reanalysis. These observations are sometimes biased (instrument or sensors malfunction,...). Are these data preprocessed before used for the comparison? If no, this could influence the realised evaluation.

In Figure 3 and associated comments in the main text, please precise if averaged variable are obtained from the respective original grid of the reanalysis, or if it's calculated after interpolation? (Please precise in the main text and in the caption.) In both case, is the spatial resolution differences could explain part of differences in the 4 variables?

There are too few assumptions to understand and explain differences between both datasets. This could help to adapt EBMs models.

*Minor Comments*

P1, L11-12 : "[…] The associated reduction in Greenland Ice Sheet's surface mass balance (SMB) leads to more runoff […] ": SMB does not lead to more runoff, but runoff leads to more negative SMB.

P1, L13: EBMs = Energy balance models and not surface mass balance models. Please clarify used acronyms.

P2, L24: ERA5 start at least in 1950. This had to be corrected everywhere else.

P2, L26: Precise SMB derived from EBMs.

P2, L54-55: Two times respective and respectively in the same sentence.

P3, L70: 1°C is inconsistent with the use of kelvin everywhere else (same in figure 1, 3 and similar figures in the supplements).

P3, L84: Please precise that the bias of 0.74 is in summer.

P4, Figure 1: Color scales are not symmetrical.

P4, Figure 2: unit is missing in subplot 2.

P6, Figure 3: I suggest to also add comparison for other surface elevation classes, at least in supplements.

---

## Author Comment (AC1)

Replies to both referees RC1 and RC2 on "Brief communication: Surface energy balance differences over Greenland between ERA5 and ERA-Interim" by U. Krebs-Kanzow et al. (EGUSPHERE-2023-525)

We found both reviewers' comments very constructive and helpful. In our response we include some new figures from additional analyses inspired by the reviewers' questions. As we see it, most of these figures are beyond the scope of the paper, but we found them quite interesting, and we offer to include them in the supplement.

Thank you very much for helping to improve the manuscript! We have compiled our replies to all raised issues and points in this document. The Referees' texts are in black print, while our responses to each item are in blue. This pdf file contains both replies in consecutively order. New figures for the manuscript and supporting figures are included at the end of this response.

**(This page is left empty intentionally.)**

**RC1**: 'Comment on egusphere-2023-525', Anonymous Referee #1, 07 Apr 2023
Review of

Brief communication: Surface energy balance differences over Greenland between ERA5 and ERA-Interim by Uta Krebs-Kanzow and others

**General**

This paper issues a warning that researchers that previously used ERA-Interim to estimate Greenland ice sheet (GrIS) surface energy/mass balance (SEB/SMB) must recalibrate their methods when switching to ERA5. The authors find significant differences in near-surface climate, notably near-surface air temperature, and SEB, notably the flux of incoming solar radiation. The paper is generally well written with clear figures but see my technical comments below. The paper is well-timed, as ERA-Interim will be phased out soon at the expense of ERA5. Unfortunately, no interpretation as to why the differences occur between the two products is provided, making the scientific impact of this study somewhat limited.

We are aware of the manuscript's descriptive nature. We wanted to disseminate the reported differences to those using ERA5 to hindcast (reproduce the historical) Greenland's surface mass balance (SMB) estimates since these differences are essential and will impact SMB estimates from surface mass and energy balance models. We have chosen the brief communication format to inform the community timely.

**Major comments**

In Figure 1 I assume some of the small-scale features in the marginal ice sheet are associated with the interpolation procedure, in combination with comparing two datasets of different resolutions.

This is most certainly right. Near the margins, changes over small spatial scales (e.g., albedo, local circulation, turbulent exchange, steep topographic gradients) will be better represented in ERA5 with roughly two times higher resolution in all spatial dimensions, and local biases are not surprising. In our manuscript, we focus on relevant differences on larger scales and which might induce systematic bias in surface mass and energy balance model results on larger scales. To clarify it, we will adjust the abstract by replacing:

"In summer, ERA5 differs significantly from ERAI, especially in the melt regions"
➔ " small scale ERA5- ERAI differences, as particularly prominent near the ice sheet's margins, can be explained by the different resolution while large scale differences might indicate a different representation of physical processes in the two reanalyses."

**Minor and textual comments**

l. 9: persistent warming trend -> persistent positive trend

We have changed it to a "persistent positive temperature trend."

l. 10: "a mean warming of 5.3 K is projected for the 21st century". This is ambiguous. By "mean" do you mean model mean, or the mean over the (remainder of) the century, or

end-of-century? Please be precise. Also, consider presenting a number from a more likely scenario.

We propose the following modification and addition: "...and an ensemble mean of SSP5-8.5 projections yields warming of 5.3 K from the first to the last two decades of the 21st century. Considering a wider range of scenarios, projections generally indicate warming over Greenland, which is weaker than *across* the *remaining Arctic*, slightly stronger than the global trends, and mostly comparable to trends over northern hemisphere land surfaces (IPCC, 2021, Fig. 4.19)".

Ref.: IPCC2021: Lee, J.-Y., J. Marotzke, G. Bala, L. Cao, S. Corti, J.P. Dunne, F. Engelbrecht, E. Fischer, J.C. Fyfe, C. Jones, A. Maycock, J. Mutemi, O. Ndiaye, S. Panickal, and T. Zhou, 2021: Future Global Climate: Scenario-Based Projections and Near- Term Information. In Climate Change 2021: The Physical Science Basis. Contribution of Working Group I to the Sixth Assessment Report of the Intergovernmental Panel on Climate Change [Masson-Delmotte, V., P. Zhai, A. Pirani, S.L. Connors, C. Péan, S. Berger, N. Caud, Y. Chen, L. Goldfarb, M.I. Gomis, M. Huang, K. Leitzell, E. Lonnoy, J.B.R. Matthews, T.K. Maycock, T. Waterfield, O. Yelekçi, R. Yu, and B. Zhou (eds.)]. Cambridge University Press, Cambridge, United Kingdom and New York, NY, USA, pp. 553–672, doi:10.1017/9781009157896.006.

l. 11: "The associated reduction in Greenland Ice Sheet's surface mass balance (SMB) leads to more runoff " It is the other way around: the associated increase in runoff leads to a reduction of the GrIS SMB...Again, please be precise, there is already enough confusion about ice sheet mass balance.

Agreed. We plan to modify as follows: "The associated increase in surface melt and runoff leads to a reduction in the GrIS SMB."

l. 13: Consider replacing "Surface mass balance models" with "Surface energy balance models".

We will use "surface mass and energy balance models".

l. 16: near-surface temperature -> near-surface air temperature

We will change it accordingly.

l. 33: Reanalysis Era-Interim (ERAI) -> ECMWF Reanalysis - Interim (ERA-Interim, henceforth ERAI)

Changed as suggested.

l. 34: Reanalysis v5 (ERA5) -> ECMWF Reanalysis v5 (ERA5)

Changed as suggested.

l. 34: ERA5 runs from January 1940 to the present

It is correct that ERA5 now dates further back to January 1940. We have corrected the text and by state "… begins in January 1940, runs until the present, ...".

l. 37: For clarity and consistency with previous work, consider using T2m rather than T2M

Will do.

l. 70: Please use higher/lower temperatures rather than warmer/colder temperatures throughout.

OK.

l. 75: stronger -> larger

Replaced as suggested.

**Citation**: https://doi.org/10.5194/egusphere-2023-525-RC1

**RC2**: 'Comment on egusphere-2023-525', Anonymous Referee #2, 09 May 2023

A Review of «Brief communication: Surface energy balance differences over Greenland between ERA5 and ERA-Interim» by Krebs-Kanzow et al.

**Overview and general comment**

Authors present here a comparison of a part of surface energy balance components between both ERA-Interim and ERA5 reanalysis with the aim to adapt SMB calculation by EBMs to the highlighted differences. Originality of the comparison come from the focus on the area below 2000m of the ice sheet to better compare the datasets over the ablation area, and the use of a temperature lapse rate to correct the 2m-temperature of differences in surface elevation when interpolated on a common 1km-grid. The comparison is clear, straightforward and well-written.

**Major Comments**

A complete analyse of the surface energy budget components (longwave radiation too) should be presented, at least in supplements if it doesn't add significant conclusions.

Thank you for this point. We agree that it is worthwhile also to consider differences in longwave radiation ($\Delta$LWD, Fig R1 lower right).

We can also demonstrate that recalculating LWD(Td, $\varepsilon$) according to Eq.1 as a function of downscaled near-surface air temperature and effective atmospheric emissivity $\varepsilon$, reduces differences in longwave radiation ($\Delta$LWDd , Fig R9, upper panel).

Finally, we can attribute differences in LWD to differences in effective temperature by considering

$\Delta$LWD$_T$ = LWD(T$_{ERA5}$, $\varepsilon_{ERA5}$) -  LWD(T$_{ERAI}$, $\varepsilon_{ERA5}$)

(Fig. R9, lower left panel).

Likewise we can identify the emissivity-related differences as

$\Delta$LWD$_e$ = LWD(T$_{ERA5}$, $\varepsilon$ERA5) -  LWD(T$_{ERA5}$, $\varepsilon$ERAI)

(Fig. R9, lower right panel). There is no indication of strong synergy between both contributors as

LWD - $\Delta$LWD$_T$ - $\Delta$LWD$_e$ << $\Delta$LWD (not shown).

To keep the paper concise, we propose to replace the cloud cover of the  original Fig. 1 with Fig. R1. We can also add  LWD in Fig. 2 and 3 (Fig R2 and Fig R3).

Emissivity is calculated on the respective grid of both reanalysis, which implies that when downscale to 1km-grid, there is no correction relative to the elevation differences whereas $\varepsilon$ is depending on the temperature. It would help to have an idea of this influence as you are considering a lapse rate to correct the temperature.

The effective atmospheric emissivity depends on temperature primarily due to the temperature dependance of the atmospheric saturation water vapor content. However,

cloud cover and atmospheric circulation are also important factors. The spatial pattern of the emissivity bias (Fig.1, lower left panel) does not indicate a strong correlation between bias and steep topographic gradients. However, inspired by your comment, we tested some first-order, linear downscaling parameters. We found that agreement between the two reanalyses and agreement with PROMICE data was reduced and not improved, while the lapse-rate correction for temperature resulted in a visible improvement. Since this paper does not aim to discuss potential downscaling strategies, we have decided not to include these results in the manuscript.

To estimate different lapse rates to correct temperature of surface elevation differences, authors calculate local lapse rates for each grid. Why don't use directly these lapse rates to correct temperature? Depending of the results, this could be add in the comparison (Figure S5) in the supplements.

As pointed out above optimizing downscaling procedures is not the focus of this paper. Nevertheless, we have included the lapse rate correction of temperature here to demonstrate that disagreement is to some extent related to resolution differences and a steep topography. Near the margins a locally diagnosed choice might indeed improve the downscaling procedure. Still, in that case, one should also consider the coefficient of determination ($R^2$) for the local linear regression between temperature and elevation, as other parameters (like distance from the coast, the surface temperature of adjacent land, etc.) might also control the temperature distribution. Fig. S 4 is intended to give an orientation, interpreting regions with homogenous slope lapse rates to indicate that a lapse rate correction is justified.

AWS data in ablation area are used to compare both reanalysis. These observations are sometimes biased (instrument or sensors malfunction,...). Are these data preprocessed before used for the comparison? If no, this could influence the realised evaluation.

For our comparison with AWS data, we utilize the clean (preprocessed) PROMICE data.

In Figure 3 and associated comments in the main text, please precise if averaged variable are obtained from the respective original grid of the reanalysis, or if it's calculated after interpolation? (Please precise in the main text and in the caption.) In both case, is the spatial resolution differences could explain part of differences in the 4 variables?

Thanks for indicating some ambiguous descriptions. We have clarified this in the text.

There are too few assumptions to understand and explain differences between both datasets. This could help to adapt EBMs models.

We will include a short discussion of which biases might be resolution dependent (and could be reduced by downscaling) and which might be related to differences in the physical parameterizations.

**Minor Comments**

P1, L11-12 : "[…] The associated reduction in Greenland Ice Sheet's surface mass balance (SMB) leads to more runoff […] ": SMB does not lead to more runoff, but runoff leads to more negative SMB.

We have rephrased this accordingly.

P1, L13: EBMs = Energy balance models and not surface mass balance models. Please clarify used acronyms.

Done; we will use "surface mass and energy balance models".

P2, L24: ERA5 start at least in 1950. This had to be corrected everywhere else.

We have corrected the starting date of ERA5 – see also reply to reviewer #1.

P2, L26: Precise SMB derived from EBMs.

OK.

P2, L54-55: Two times respective and respectively in the same sentence.

We have removed one.

P3, L70: 1°C is inconsistent with the use of kelvin everywhere else (same in figure 1, 3 and similar figures in the supplements).

Thanks for indicating this inconsistent use of the units. We use the Kelvin unit ("K") on page three, line 70 (P3, L70) and page four, line 94 (P4, L94).

P3, L84: Please precise that the bias of 0.74 is in summer.

OK

P4, Figure 1: Color scales are not symmetrical.

The upper and lower ends of the colorbar reflect the non-symmetric value range. Otherwise, the colorbar is symmetric.

P4, Figure 2: unit is missing in subplot 2.

Thanks for reporting the missing units. We have added the missing unit information to the related property label of the x-axis.

P6, Figure 3: I suggest to also add comparison for other surface elevation classes, at least in supplements.

We have done so for intervals [0,1000], [1000,2000],[2000,3000],[3000,4000] : (Fig. R4 - Fig.R7).

[Figure]

**Figure R1.** (This Figure could replace Fig. 1 in the article) Mean bias between ERA5 and ERAI for the summer mean (i.e., June, July, and August, JJA) 1979–2018 period of the 2m-air temperature (top left), downward shortwave radiation (top right), emissivity (bottom left) and downward longwave radiation (bottom right). Stippling indicates regions where the mean bias is smaller than two respective standard deviations.

**Figures**

[Figure]

**Figure R2.** (The second panel could be added to Fig. 2 in the article) Distribution of ERA5 and ERAI biases with respect to monthly PROMICE observations for the summer months (June, July, August) in 2007–2016: downward longwave radiation (top), downward longwave radiation as recalculated via lapse rate corrected temperatures and emissivity The text box insets provide standard deviation (std) and mean biases (MB) for the respective distributions.

[Figure]

**Figure R3.** (This Figure could replace Fig. 3 in the article) 1979–2016 yearly summer means of (a) 2m-air temperature, (b) downward surface shortwave radiation, (c) emissivity, and (d) cloud cover for ERA5 (blue) and ERAI (red) averaged over the 0 m to 2000 m elevation range of the GrIS.

[Figure]

**Figure R4.** (This Figure could be added to the supplement) 1979–2016 yearly summer means of (a) 2m-air temperature, (b) downward surface shortwave radiation, (c) emissivity, and (d) cloud cover for ERA5 (blue) and ERAI (red) averaged over the 0 m to 1000 m elevation range of the GrIS.

[Figure]

**Figure R5.** (This Figure could be added to the supplement) 1979–2016 yearly summer means of (a) 2m-air temperature, (b) downward surface shortwave radiation, (c) emissivity, and (d) cloud cover for ERA5 (blue) and ERAI (red) averaged over the $1000\,\mathrm{m}$ to $2000\,\mathrm{m}$ elevation range of the GrIS.

[Figure]

**Figure R6.** (This Figure could be added to the supplement) 1979–2016 yearly summer means of (a) 2m-air temperature, (b) downward surface shortwave radiation, (c) emissivity, and (d) cloud cover for ERA5 (blue) and ERAI (red) averaged over the 2000 m to 3000 m elevation range of the GrIS.

[Figure]

**Figure R7.** (This Figure could be added to the supplement) 1979–2016 yearly summer means of (a) 2m-air temperature, (b) downward surface shortwave radiation, (c) emissivity, and (d) cloud cover for ERA5 (blue) and ERAI (red) averaged over the 3000 m to 4000 m elevation range of the GrIS.

[Figure]

**Figure R8.** (This Figure could replace Fig.S3 in the Supplement) Same as Fig. 1 but for annual means.

[Figure]

**Figure R9.** Mean bias between ERA5 and ERAI for the summer mean (i.e., June, July, and August, JJA) 1979–2018 period of the downward longwave radiation, origional LWD (top left), as recalculated via lapse rate corrected temperatures and emissivity (top right), temperature related bias contribution (bottom left, recalculated based on same emissivity but ERA5 and ERAI temperature), emissivity related bias contribution (bottom right, recalculated based on same temperature but ERA5 and ERAI emissivity)

---

## Author Response (AR1)

Replies to both referees RC1 and RC2 on "Brief communication: Surface energy balance differences over Greenland between ERA5 and ERA-Interim" by U. Krebs-Kanzow et al. (EGUSPHERE-2023-525)

We found both reviewers' comments very constructive and helpful.

Thank you very much for helping to improve the manuscript! We have compiled our replies to all raised issues and points in this document. A point-by-point response is given below. The Referees' texts are in black print, while our responses to each item are in blue. Line numbers refer to the document with highlighted changes. This pdf file contains both replies in consecutively order.

**(This page is left empty intentionally.)**

**RC1**: 'Comment on egusphere-2023-525', Anonymous Referee #1, 07 Apr 2023
Review of

Brief communication: Surface energy balance differences over Greenland between ERA5 and ERA-Interim by Uta Krebs-Kanzow and others

**General**

This paper issues a warning that researchers that previously used ERA-Interim to estimate Greenland ice sheet (GrIS) surface energy/mass balance (SEB/SMB) must recalibrate their methods when switching to ERA5. The authors find significant differences in near-surface climate, notably near-surface air temperature, and SEB, notably the flux of incoming solar radiation. The paper is generally well written with clear figures but see my technical comments below. The paper is well-timed, as ERA-Interim will be phased out soon at the expense of ERA5. Unfortunately, no interpretation as to why the differences occur between the two products is provided, making the scientific impact of this study somewhat limited.

We are aware of the manuscript's descriptive nature. We wanted to disseminate the reported differences to those using ERA5 to hindcast (reproduce the historical) Greenland's surface mass balance (SMB) estimates since these differences are essential and will impact SMB estimates from surface mass and energy balance models. We have chosen the brief communication format to inform the community timely.

**Major comments**

In Figure 1 I assume some of the small-scale features in the marginal ice sheet are associated with the interpolation procedure, in combination with comparing two datasets of different resolutions.

This is most certainly right. Near the margins, changes over small spatial scales (e.g., albedo, local circulation, turbulent exchange, steep topographic gradients) will be better represented in ERA5 with roughly two times higher resolution in all spatial dimensions, and local biases are not surprising. In our manuscript, we focus on relevant differences on larger scales and biases which might induce systematic bias in surface mass and energy balance model results on larger scales. We have added some lines in the abstract (l. 3ff) and conclusion(l. 133ff).

**Minor and textual comments**

l. 9: persistent warming trend -> persistent positive trend

We have changed it to a "persistent positive temperature trend."

l. 10: "a mean warming of 5.3 K is projected for the 21st century". This is ambiguous. By "mean" do you mean model mean, or the mean over the (remainder of) the century, or end-of-century? Please be precise. Also, consider presenting a number from a more likely scenario.

We have made the following modification (l.12ff): "...and an ensemble mean of SSP5-8.5 projections yields warming of 5.3 K from the first to the last two decades of the 21st century. Considering a wider range of scenarios, projections generally

indicate warming over Greenland*, which is weaker than across the remaining Arctic*, slightly stronger than the global trends*, and mostly comparable to trends over northern hemisphere land surfaces (IPCC, 2021, Fig. 4.19)".

Ref.: IPCC2021: Lee, J.-Y., J. Marotzke, G. Bala, L. Cao, S. Corti, J.P. Dunne, F. Engelbrecht, E. Fischer, J.C. Fyfe, C. Jones, A. Maycock, J. Mutemi, O. Ndiaye, S. Panickal, and T. Zhou, 2021: Future Global Climate: Scenario-Based Projections and Near- Term Information. In Climate Change 2021: The Physical Science Basis. Contribution of Working Group I to the Sixth Assessment Report of the Intergovernmental Panel on Climate Change [Masson-Delmotte, V., P. Zhai, A. Pirani, S.L. Connors, C. Péan, S. Berger, N. Caud, Y. Chen, L. Goldfarb, M.I. Gomis, M. Huang, K. Leitzell, E. Lonnoy, J.B.R. Matthews, T.K. Maycock, T. Waterfield, O. Yelekçi, R. Yu, and B. Zhou (eds.)]. Cambridge University Press, Cambridge, United Kingdom and New York, NY, USA, pp. 553–672, doi:10.1017/9781009157896.006.

l. 11: "The associated reduction in Greenland Ice Sheet's surface mass balance (SMB) leads to more runoff " It is the other way around: the associated increase in runoff leads to a reduction of the GrIS SMB...Again, please be precise, there is already enough confusion about ice sheet mass balance.

Agreed. We have modified this (l.17): "The associated increase in surface melt and runoff leads to a reduction in the GrIS SMB."

l. 13: Consider replacing "Surface mass balance models" with "Surface energy balance models".

We now use "surface mass and energy balance models".

l. 16: near-surface temperature -> near-surface air temperature

We have changed it accordingly.

l. 33: Reanalysis Era-Interim (ERAI) -> ECMWF Reanalysis - Interim (ERA-Interim, henceforth ERAI)

Changed as suggested.

l. 34: Reanalysis v5 (ERA5) -> ECMWF Reanalysis v5 (ERA5)

Changed as suggested.

l. 34: ERA5 runs from January 1940 to the present

It is correct that ERA5 now dates further back to January 1940. We have corrected the text and by state "… begins in January 1940, runs until the present, ...".

l. 37: For clarity and consistency with previous work, consider using T2m rather than T2M

done.

l. 70: Please use higher/lower temperatures rather than warmer/colder temperatures throughout.

OK.

l. 75: stronger -> larger

Replaced as suggested.

**Citation**: https://doi.org/10.5194/egusphere-2023-525-RC1

A Review of «Brief communication: Surface energy balance differences over Greenland between ERA5 and ERA-Interim» by Krebs-Kanzow et al.

**Overview and general comment**

Authors present here a comparison of a part of surface energy balance components between both ERA-Interim and ERA5 reanalysis with the aim to adapt SMB calculation by EBMs to the highlighted differences. Originality of the comparison come from the focus on the area below 2000m of the ice sheet to better compare the datasets over the ablation area, and the use of a temperature lapse rate to correct the 2m-temperature of differences in surface elevation when interpolated on a common 1km-grid. The comparison is clear, straightforward and well-written.

**Major Comments**

A complete analyse of the surface energy budget components (longwave radiation too) should be presented, at least in supplements if it doesn't add significant conclusions.

Thank you for this point. We agree that it is worthwhile also to consider differences in longwave radiation and added longwave radiation to the analysis and now consider cloud cover only in the supplement (Fig. S8) .

Emissivity is calculated on the respective grid of both reanalysis, which implies that when downscale to 1km-grid, there is no correction relative to the elevation differences whereas ε is depending on the temperature. It would help to have an idea of this influence as you are considering a lapse rate to correct the temperature.

The effective atmospheric emissivity depends on temperature primarily due to the temperature dependance of the atmospheric saturation water vapor content. However, cloud cover and atmospheric circulation are also important factors. The spatial pattern of the emissivity bias (Fig.1, lower left panel) does not indicate a strong correlation between bias and steep topographic gradients. However, inspired by your comment, we tested some first-order, linear downscaling parameters. We found that agreement between the two reanalyses and agreement with PROMICE data was reduced and not improved, while recalculating longwave radiation as a function of emissivity and lapse-rate corrected temperature resulted in improved statistics (Supplement Fig. S9). Since this paper does not aim to discuss potential downscaling strategies, we have decided to include these results in the supplement.

To estimate different lapse rates to correct temperature of surface elevation differences, authors calculate local lapse rates for each grid. Why don't use directly these lapse rates to correct temperature? Depending of the results, this could be add in the comparison (Figure S5) in the supplements.

Optimizing downscaling procedures is not the focus of this paper. Nevertheless, we have included the lapse rate correction of temperature here to demonstrate that

disagreement is to some extent related to resolution differences and a steep topography. Near the margins a locally diagnosed choice might indeed improve the downscaling procedure. Still, in that case, one should also consider the coefficient of determination ($R^2$) for the local linear regression between temperature and elevation, as other parameters (like distance from the coast, the surface temperature of adjacent land, etc.) might also control the temperature distribution. Fig. S 4 is intended to give an orientation, interpreting regions with homogenous slope lapse rates to indicate that a lapse rate correction is justified.

AWS data in ablation area are used to compare both reanalysis. These observations are sometimes biased (instrument or sensors malfunction,...). Are these data preprocessed before used for the comparison? If no, this could influence the realised evaluation.

For our comparison with AWS data, we utilize the clean (preprocessed) PROMICE data.

In Figure 3 and associated comments in the main text, please precise if averaged variable are obtained from the respective original grid of the reanalysis, or if it's calculated after interpolation? (Please precise in the main text and in the caption.) In both case, is the spatial resolution differences could explain part of differences in the 4 variables?

We have clarified this in the figure caption.

There are too few assumptions to understand and explain differences between both datasets. This could help to adapt EBMs models.

We have extended the conclusion with a discussion of which biases might be resolution dependent (and could be reduced by downscaling) and which might be related to differences in the physical parameterizations.

**Minor Comments**
P1, L11-12 : "[…] The associated reduction in Greenland Ice Sheet's surface mass balance (SMB) leads to more runoff […] ": SMB does not lead to more runoff, but runoff leads to more negative SMB.

We have rephrased this accordingly.

P1, L13: EBMs = Energy balance models and not surface mass balance models. Please clarify used acronyms.

Done; we use "surface mass and energy balance models".

P2, L24: ERA5 start at least in 1950. This had to be corrected everywhere else.

We have corrected the starting date of ERA5 – see also reply to reviewer #1.

P2, L26: Precise SMB derived from EBMs.

OK.

P2, L54-55: Two times respective and respectively in the same sentence.

We have removed one.

P3, L70: 1°C is inconsistent with the use of kelvin everywhere else (same in figure 1, 3 and similar figures in the supplements).

Done.

P3, L84: Please precise that the bias of 0.74 is in summer.

OK

P4, Figure 1: Color scales are not symmetrical.

The upper and lower ends of the colorbar reflect the non-symmetric value range. Otherwise, the colorbar is symmetric.

P4, Figure 2: unit is missing in subplot 2.

We have added the missing unit information to the label of the x-axis.

P6, Figure 3: I suggest to also add comparison for other surface elevation classes, at least in supplements.

We have done so for intervals [0,1000], [1000,2000],[2000,3000],[3000,4000] : (Supplement Figs. S10-S13).

---

## Author Response (AR2)

Dear Editor,
thank you for your comments. We followed your suggestions and are submitting an accordingly modified new version. We also corrected one typo (L. 91 "rare" -> rate) and added some commas.

L17: sea level rise => sea-level rise
OK
L35: focussing => focusing
OK
L46: T2m <= Consider using subscript "T_2m", or removing italics. I assume this name relates directly to its name in the ECMWF dataset. Check for consistency throughout the manuscript.
OK
L48, Eq. 1: Should the "int" subscript appear here? This is a general equation used for both reanalyses right?
indeed the subscript "int" was referring to "interpolated" here, but it was wrong anyway, as we calculate the emissivity from coarse resolution. Thank you for spotting this.
L55: "ISMIP6 orography H_ice" <= Are temperatures scaled by H_ice or surface elevation, as the latter would be the more relevant variable. On line 60, surface elevation is even mentioned. Usually H = ice thickness and h = surface elevation (or z_srf, for example would be more direct). Please just double check that you are happy with this choice.
good point, we changed H_ice to h_ismip and H_int to h_int
L75: South Eastern margins = > southeastern margins
OK
L80 and rest of section: the term "mean bias" is somewhat confusing in this context, as bias is usually relative to a chosen reference. Here you are comparing ERAI to ERA5 and it is not clear which one you assume then to be the reference. This is also problematic when you refer to the two standard deviations ("The mean bias exceeds two standard deviations") as again the question arises as to which dataset provides the measure of the standard deviation. To solve this, I would add a few sentences to define your methlodogy more clearly. State explicitly which dataset is considered the reference. Or, if you don't mean to define a reference, then change from "bias" to simply "difference". In this case, it should be made clear how the standard deviation is calculated that is used as the measure of variability. Note that when comparing to the AWS data, I see no problem, as clearly the AWS data would be the reference and the renalysis is biased relative to that.
we agree and replaced bias with differences and reformulated the respective parts.
L92: comparrison = > comparison
OK
L94: Fig. 2) => Figure 2 [When starting a setence, the whole word should be used.]
OK
L99: In Fig. 3 height range => In Fig. 3, the elevation range considered
OK
L124: In case => In the case
OK
L124: air temperature => air temperature,
OK